# Current Trends and Controversies in the Management of Warthin Tumor of the Parotid Gland [note 1]

**DOI:** 10.3390/diagnostics11081467

**Published:** 2021-08-13

**Authors:** Miquel Quer, Juan C. Hernandez-Prera, Carl E. Silver, Maria Casasayas, Ricard Simo, Vincent Vander Poorten, Orlando Guntinas-Lichius, Patrick J. Bradley, Wai Tong-Ng, Juan P. Rodrigo, Antti A. Mäkitie, Alessandra Rinaldo, Luiz P. Kowalski, Alvaro Sanabria, Remco de Bree, Robert P. Takes, Fernando López, Kerry D. Olsen, Ashok R. Shaha, Alfio Ferlito

**Affiliations:** 1Department of Otorhinolaryngology, Hospital Santa Creu I Sant Pau, Universitat Autònoma de Barcelona, 08041 Barcelona, Spain; mcasasayas@santpau.cat; 2Multidisciplinary Salivary Gland Society, 1204 Genève, Switzerland; vincent.vanderpoorten@uzleuven.be (V.V.P.); orlando.guntinas@med.uni-jena.de (O.G.-L.); pjbradley@zoo.co.uk (P.J.B.); 3Department of Pathology, Moffitt Cancer Center, Tampa, FL 33612, USA; Juan.Hernandez-Prera@moffitt.org; 4Department of Surgery, University of Arizona College of Medicine, Phoenix, AZ 85721, USA; csilver@cox.net; 5Department of Otorhinolaryngology Head and Neck Surgery, Guy’s and St Thomas’ Hospital NHS Foundation Trust, London SE1 9RT, UK; Ricard.Simo@gstt.nhs.uk; 6Otorhinolaryngology-Head and Neck Surgery, University Hospitals Leuven, 3000 Leuven, Belgium; 7Department of Oncology, Section Head and Neck Oncology, KU Leuven, 3000 Leuven, Belgium; 8Department of Otorhinolaryngology, Institute of Phoniatry/Pedaudiology, Jena University Hospital, 07740 Jena, Germany; 9Facial Nerve Center, Jena University Hospital, 07740 Jena, Germany; 10Department of Otolaryngology, Head and Neck Surgery, Nottingham University Hospital NHS Trust, Queens Medical Centre Campus, Nottingham NG7 2UH, UK; 11Department of Clinical Oncology, LKS Faculty of Medicine, The University of Hong Kong, Hong Kong 999077, China; ngwt1@hku.hk; 12Department of Otorhinolaryngology, Head and Neck Surgery, Hospital Universitario Central de Asturias, 33011 Oviedo, Spain; jprodrigo@uniovi.es (J.P.R.); fernandolopezphd@gmail.com (F.L.); 13Instituto de Investigación Sanitaria del Principado de Asturias (ISPA), Instituto Universitario de Oncología del Principado de Asturias (IUOPA), University of Oviedo CIBERONC-ISCIII, 33011 Oviedo, Spain; 14Department of Otorhinolaryngology-Head and Neck Surgery, University of Helsinki and Helsinki University Hospital, 00029 HUS Helsinki, Finland; Antti.Makitie@hus.fi; 15Division of Ear, Nose and Throat Diseases, Department of Clinical Sciences, Intervention and Technology, Karolinska University Hospital, 141 86 Stockholm, Sweden; 16University of Udine School of Medicine, 33100 Udine, Italy; alessandra.rinaldo@uniud.it; 17Head and Neck Surgery Department, University of Sao Paulo Medical School, São Paulo 01509, Brazil; lp_kowalski@uol.com.br; 18Head and Neck Surgery and Otorhinolaryngology Department, A.C. Camargo Cancer Center, São Paulo 01509, Brazil; 19Department of Surgery, School of Medicine, Universidad de Antioquia, IPS Universitaria-Hospital Universitario San Vicente Fundación, Medellin 050010, Colombia; alvarosanabria@gmail.com; 20Department of Head and Neck Surgical Oncology, University Medical Center Utrecht, 3584 CX Utrecht, The Netherlands; R.deBree@umcutrecht.nl; 21Department of Otolaryngology-Head and Neck Surgery, Radboud University Medical Center, 6525 GA Nijmegen, The Netherlands; Robert.Takes@radboudumc.nl; 22Department of Otorhinolaryngology, Mayo Clinic, Rochester, MN 55902, USA; olsen.kerry@mayo.edu; 23Head and Neck Service, Memorial Sloan-Kettering Cancer Center, New York, NY 10065, USA; shahaa@mskcc.org; 24Coordinator of the International Head and Neck Scientific Group, 35125 Padua, Italy; profalfioferlito@gmail.com

**Keywords:** Warthin tumor, papillary cystadenoma lymphomatosum, cystadenolymphoma, adenolymphoma, lymphomatous adenoma, parotid tumor

## Abstract

Purpose: To review the current options in the management of Warthin tumors (WTs) and to propose a working management protocol. Methods: A systematic literature search was conducted using PubMed and ScienceDirect database. A total of 141 publications were selected and have been included in this review. Publications were selected based on relevance, scientific evidence, and actuality. Results: The importance of parotid WTs is increasing due to its rising incidence in many countries, becoming the most frequently encountered benign parotid tumor in certain parts of the world. In the past, all WTs were treated with surgery, but because of their slow growth rate, often minimal clinical symptoms, and the advanced age of many patients, active observation has gradually become more widely used. In order to decide on active surveillance, the diagnosis of WT must be reliable, and clinical, imaging, and cytological data should be concordant. There are four clear indications for upfront surgery: uncertain diagnosis; cosmetic problems; clinical complaints, such as pain, ulceration, or recurrent infection; and the patient’s wish to have the tumor removed. In the remaining cases, surgery can be elective. Active surveillance is often suggested as the first approach, with surgery being considered if the tumor progresses and/or causes clinical complaints. The extent of surgery is another controversial topic, and the current trend is to minimize the resection using partial parotidectomies and extracapsular dissections when possible. Recently, non-surgical options such as microwave ablation, radiofrequency ablation, and ultrasound-guided ethanol sclerotherapy have been proposed for selected cases. Conclusions: The management of WT is gradually shifting from superficial or total parotidectomy to more conservative approaches, with more limited resections, and to active surveillance in an increasing number of patients. Additionally, non-surgical treatments are emerging, but their role needs to be defined in future studies.

## 1. Introduction

The Warthin tumor (WT) has been identified as the second most frequent benign tumor of the parotid gland, following pleomorphic adenoma [1]. WT is most frequently diagnosed in middle-aged or elderly patients, especially in those who smoke (>50 years), with a male preponderance. It is usually situated at the inferior pole of the parotid gland, level II in the European Salivary Gland Classification [2]. WT is an adenoma composed of a bilayered columnar and basaloid oncocytic epithelium that forms multiple cysts with numerous papillae, accompanied by a proliferation of follicle-containing lymphoid tissue [1]. The synonyms for WT include papillary cystadenoma lymphomatosum, adenolymphoma, and lymphomatous adenoma [1].

WTs usually present as a slowly growing, painless, or mildly uncomfortable swelling or mass on the inferior pole of the parotid gland, which often feels soft. Around 15% of WTs are multifocal in the same gland and in approximately 10% of cases are present bilaterally [1,3]. WTs have been reported in association with other salivary gland neoplasms (benign and malignant) [4]. It has also been reported that WTs arise in intra- and peri-parotid lymph nodes and may account for the diagnosis of the WT-isolated extraparotid gland in the neck [1,5].

The most common treatment has been a superficial or partial superficial parotidectomy. However, there still exist controversies on the extent of the surgery and in recent years extracapsular dissection is increasingly used. In fact, when a tumor is found during the follow-up, it is likely attributed to a second tumor rather than a true recurrence due to inadequate resection, given the fact that a WT may be multifocal [6]. Furthermore, the malignant transformation of a WT is extremely rare [1]. In recent years, active surveillance [7] has been proposed, but there is still no clear consensus on exactly which to choose for observation versus surgery. In this paper, the most recent literature is summarized in a systematic review and a general management guideline is proposed.

## 2. Methods

A scoping review of the current literature was conducted using PubMed and ScienceDirect databases from 1980 to 2021, with the last update on 18 May 2021, for English language publications using the following search criteria in the title or abstract: “parotid”, coupled with “Warthin’s tumor”, “Warthin tumor”, “Warthin’s tumor”, “Warthin tumor”, or “Cystadenolymphoma”. The search strategy aimed to identify publications that discuss treatment of WTs. When there was information in the abstract addressing WT treatment, the full-text article was reviewed. References from any full-text articles were cross-checked to ensure the inclusion of all relevant publications (Figure 1). The results were then manually filtered by using the following criteria: (1) WT of the parotid gland, (2) information on the prevalence of WT (3) information on diagnosis, and (4) information on treatment options.

The preliminary literature search identified 581 publications. After sorting and removal of duplicates, 141 papers were retrieved and reviewed in detail. The highest evidence level reached was retrospective comparative studies (evidence-based medicine level III). However, most studies were retrospective case series (level IV). The 141 studies included a total of 8649 patients with WT of the parotid gland.

Given the restraints in the number of references allowed by the journal, only 50 references are presented within the review. The 141 articles selected for the manuscript can be found in Appendix A.

## 3. Epidemiology

The diagnosis and, hence, the incidence of WTs has steadily increased since the original description of Aldred Scoot Warthin (USA) in 1929 [8]. He wrote “In a diagnostic material of several hundred thousand cases examined by me since 1895, there have been over seven hundred mixed tumors of the parotid gland, and over five hundred branchial cyst of the cervical region, and two cases only of a form of tumor of the parotid region, which I must believe to be of very great rarity, not only because of its extremely low incidence in my material, but also because of the silence in the literature regarding it”.

So, the WT was rarely reported until at least 1929, but slowly has become the second most prevalent benign tumor and accounts for 4% to 25% of all salivary gland tumors [1]. Nearly all WTs arise in the parotid gland. Most involve the tail of the gland, but about 10% occur in the deep lobe. Rare examples have been reported at sites other than the parotid gland. Between 10% and 27% of patients develop more than one WT; these are bilateral in 5% to 30% of patients. The average age at presentation is around 60 years and there is a male predominance with a male:female ratio around 1.5:1. Fewer than 6% of WTs occur before 40 years age [1,9].

Recently, three studies from Germany described a rising incidence of WTs [10,11,12]. Luers et al. [10] described the rising relative incidence of WTs in the past 25 years in comparison to pleomorphic adenoma, showing an increase of WTs from 24% of all parotid gland tumors in 1990 to 48% in 2014. Franzen et al. [11] reported that WTs increased steadily from 20.6% in the decade between 1975 and 1986 to 44.9% in the decade between 2008 and 2017, and moreover after 1996, WTs turned out to be the most frequent tumor entity. Additionally, this study showed that the male-to-female ratio in WTs decreased significantly (from 5.3 to 1 down to 2.1 to 1, respectively) and that the percentage of smokers with WTs remained stable and was 92% in total. Finally, Psychogios et al. [12] found that WTs were the most common benign lesion in their series of surgical parotid cases, in which 201 cases were WTs (42.4%) followed by 138 pleomorphic adenomas (29.1%). Additional publications from Poland [13] and Turkey [14] reported similar findings.

Different explanations have been proposed for this rising incidence. The most common explanation is the increase of tobacco consumption. Smokers have eight times more risk of developing WTs compared with non-smokers. As tobacco use increased from 1920 to 1980 in Europe, this can probably explain part of the incidence growth in the period from Warthin′s original description. However, a retrospective study from Austria by Kadletz [15] demonstrated that tobacco use decreased in recent years and the incidence of WTs continues to increase, which may imply other etiological factors. Those authors suggested that a high body mass index (BMI) and metabolic syndrome may increase the risk of WT formation. The investigators found a higher BMI in patients with a WT compared with patients with other benign parotid tumors. In addition, these patients presented more metabolic syndrome-associated comorbidities compared with patients with other benign parotid tumors.

Another possible explanation is that due to the increasing use of imaging studies in daily practice in recent years (MRI, PET, etc.) there are more diagnoses of WTs in asymptomatic patients that would not have been otherwise diagnosed (incidentalomas). As a matter of fact, a significant number of WTs present as incidentalomas, especially in positron emission computed tomography [16]. Due to high FDG uptake, these WTs are easily diagnosed even if FDG-PET/CT is made for disease outside the head and neck area. Therefore, it is debatable if there is an actual increase of WTs or that it is due to rise of incidentalomas, which are likely to be more frequently diagnosed in recent decades because of improvements and the increasing use of imaging techniques.

In any case, the increase in its incidence in combination with a significant number of asymptomatic cases makes the debate on optimal management truly relevant.

## 4. Diagnosis of Warthin Tumors (WT)

Three aspects must be evaluated as follows: (1) clinical data, (2) imaging, and (3) cytology.

The most important clinical information includes age (usually more than 50 years old), smoking history, and location of the tumor (superficial, level II in the majority of cases [17]). In addition, bilateral, or multiple tumors are characteristic of WTs, as well as a previous homo or contralateral WT diagnosis. Patients under 40 years of age or with a solitary deep lobe tumor should be suspect and investigated for other diagnoses.

Imaging is especially important and can certainly help in the diagnosis of a WT (references in Appendix A). There are several available imaging modalities that we will not address in this manuscript; however, ultrasound, CT, MRI, and PET-CT are reliable imaging techniques and radiomics may be helpful in the diagnosis of WTs. Depending on the availability and experience of each center, a choice has to be made of which technique adds most to the diagnosis. Finally, cytology and or core needle biopsy should be concordant with the clinical and radiological diagnosis of WTs.

The accuracy of the diagnosis is based on the concordance of the three modalities: clinical, imaging, and cytology evaluation. If it is not feasible to reach a WT diagnosis, surgery should be indicated to assure the definitive diagnosis by histopathological examination of the resected specimen. When the diagnosis of a WT can reliably be established, optimal management may be debated. Surgery is not mandatory for all cases, and active surveillance can be an alternative or first choice (Figure 2).

## 5. Diagnostic Pathology

Overall, the cytological, macroscopic, and microscopic characteristics of a WT are typically straightforward, and, in most cases, does not represent a diagnostic challenge.

A fine needle aspiration biopsy (FNAB) identifying the triad of small oncocytic cells arranged in cohesive sheets or in papillary structures, numerous polymorphic lymphocytes, and a granular proteinaceous “dirty” background is used for the diagnostic [18]. In contrast to a surgical pathology specimen, cytology smears may be less representative, and the differential diagnosis in a FNAB is contingent to the combination of these three cytological components. Finding oncocytes and lymphocytes is common and, in this scenario, the diagnosis can easily be established. In contrast, when only one element is present, diagnostic difficulties may arise. The identification of only oncocytes may suggest various oncocytoid salivary gland neoplasms (benign or malignant), as the presence of lymphocytes alone might be confused with an intraparotid lymph node or lymphadenitis, while a salivary duct cyst can be suspected if the background granular debris predominates. As discussed below, squamous and mucous cell metaplasia are important diagnostic pitfalls in the WT, and in a cytology specimen, the identification of these findings might lead to the diagnosis of a squamous cell carcinoma or mucoepidermoid carcinoma. In addition, a granular “dirty” background can, on occasion, mimic the necrotic background characteristic of carcinomas [19].

The Milan System for Reporting Salivary Gland Cytopathology is gaining popularity and, following this reporting schema, WTs showing all these components fall within the diagnostic Category IV.A “Benign Neoplasm”, while oncocytic-predominant tumors might be classified as Category IV.B “Salivary Gland Neoplasm with Uncertain Malignant Potential (SUMP)” [20]. A recent international and multi-institutional study demonstrated an extremely high diagnostic accuracy of 96.1% for the diagnosis of WTs when this system is applied. The risk of malignancy in the benign neoplasm category for cases diagnosed as WTs is 1.3%; however, it is important to recognized that the risk of malignancy in the SUMP category for cases with a differential of WT was statistically higher (18.8%). This marked and clinically important variation in the risk of malignancy corresponds to broad differential diagnosis of oncocytoid salivary gland neoplasms [21].

Macroscopically, a WT is an encapsulated, soft, and fluctuant, round to oval mass. Not uncommonly, lesions can be multiple. The cut surface is tan-brown, smooth, or lobulated with multiple cystic spaces containing a brown to green viscous fluid. Microscopically at scanning magnification, the tumors are well-circumscribed, exhibit variable papillary and cystic architecture, and show a combination of oncocytic epithelial elements and lymphoid stroma (Figure 2). Some lesions occur within the intra- and periparotid parotid lymph nodes, while others are centered within the salivary gland parenchyma. The oncocytic epithelium that lines the papillae and cystic structures is bilayered. The inner luminal cells are tall and columnar with nuclei polarized towards the lumen, while outer basal cells have a cuboidal/polygonal configuration with basally located nuclei. The cytoplasm of both cell layers is densely granular and eosinophilic, while the nuclei are round to oval with conspicuous nucleoli. Scattered ciliated cells, mucocytes, as well as sebaceous cells can be appreciated. The lymphoid stroma is sharply demarcated from the epithelial component and is composed of mature lymphocytes showing well-formed lymphoid follicles with germinal centers. Usually, the lymphoid stroma is evenly distributed throughout the tumor; however, lymphoid stroma-poor lesions exist. Thick granular eosinophilic secretions are present with lumen of the cystic spaces and within the papillae.

WTs can undergo infarction, either spontaneously or after a traumatic event such as FNAB, and when this phenomenon occurs, the term metaplastic WT is used [22]. Microscopically, metaplastic WTs show various degrees of necrosis with ghost-like papillae, which can be highlighted with keratin or reticulin staining. Adjacent to necrotic areas, squamous and mucocyte metaplasia can be discerned. Reactive stromal changes associated with acute and chronic inflammation and granulation tissue may also be present. In many cases, the correct diagnosis is established by identifying a residual focus of a non-infarcted WT. However, when the squamous and mucocyte metaplasia is extensive, a WT may be misdiagnosed as a squamous cell carcinoma or mucoepidermoid carcinoma. In such cases, the lack of infiltrative growth and absences of marked pleomorphism and atypical mitosis in the squamous component supports a metaplastic process. In difficult cases where the diagnosis mucoepidermoid carcinoma is entertained, the evaluation for *MAML2* gene rearrangement can solve the question, as a metaplastic WT does not harbor this molecular alteration [23].

On the other hand, a variant mucoepidermoid carcinoma—the so-called Warthin-like mucoepidermoid carcinoma—may be particularly problematic and must be distinguished from a WT [24]. Histologically, a Warthin-like mucoepidermoid carcinoma consists of multiloculated cystic structures associated with prominent lymphoid stroma including germinal centers. Cystic structures show a multilayered epithelium with variable eosinophilic to oncocytic cells with scattered mucocytes. Microscopic foci of the conventional mucoepidermoid carcinoma exhibiting epidermoid and intermediate cells are often present and their identification aids in making the current diagnosis. In equivocal cases, the presence of a *MAML2* rearrangement confirms the malignant diagnosis [24,25]. Another differential diagnosis of a WT includes oncocytoma and nodular oncocytic hyperplasia. The former consists of a well-circumscribed lesion composed of oncocytic cells arranged in solid, trabecular, cord-like, or organoid growth patterns separated by a thin fibroconnective tissue stroma, while the latter is characterized by unencapsulated irregular oncocytic proliferation with solid, tubular, or trabecular patterns that are randomly distributed throughout the salivary parenchyma [26,27]. None of these lesions are typically associated with a rich lymphoid stroma.

## 6. Primary Surgical Excision

Surgery currently remains the primary recommended treatment of WTs. The extent of the surgery depends on the size and location of the tumor [28,29]. Based on our evaluation of the literature and personal experience, there are four clear indications for upfront surgery (Table 1): (1) no reliable or no conclusive diagnosis of WT; (2) cosmetic concerns; (3) clinical complaints such as pain, infection, or ulceration [30] and (4) the patient’s wishes to have the tumor removed. In other cases, surgery can be an elective choice, and active surveillance may be the first approach (Figure 3).

Surgery is most frequently recommended when WT causes cosmetic concerns and objective clinical deformity is the patient′s primary complaint and concern (Figure 4). Other clinical issues, mainly pain, recurrent infection, or ulceration (Figure 5) may also require surgical treatment. On occasion, the patient may present with urgency at the emergency department with acute swelling and fever suggesting an infection or abscess formation, which requires aspiration diagnosis and a course of antibiotics [29,31]. Pain can be present in some WT patients, especially when the tumor is compressed. However, with regard to pain, one should keep in mind that the tumor may represent another type of salivary gland tumor or could be malignant, which is more likely to cause this symptom [4]. Finally, some patients clearly prefer to have their WT removed because of persistent fear of an underlying malignancy.

## 7. Extent of Surgery

Extent of resection, varying from minimal surgeries to reduce postoperative complications and sequelae, to more extensive surgical procedures to lower the possibility of recurrences or metachronous WTs, is still controversial. Attempts have been made at the classification of parotid tumors in terms of size and location to aid with determining the extent of surgical excision [28,29], but there is no consensus on this matter. In Table 2, a general guideline for surgical approach in various clinical settings is proposed.

### 7.1. Parotidectomy I–IV (Total Parotidectomy)

Although in the past some authors [32] did recommend total parotidectomy as the modality of choice for all WTs due to the high incidence of multifocality, currently this option is mainly proposed for those rare cases with multiple WTs additionally involving the deep lobe [33] or for cases with multiple WTs when the surgeon wants to minimize the risk of occult deep lobe lesions [34]. The advantage of total parotidectomy in multifocal WTs is the ability to resect all possible foci and to minimize the risk of second ipsilateral tumors. The disadvantage is an increase in occurrence of potential postoperative morbidity (facial nerve palsy, Frey′s syndrome, poor cosmetic result), especially in elderly patients.

Even in the case of multiple WTs, Mantsopoulos et al. [35] compared total parotidectomy to multiple extracapsular dissection (ECD) of the individual tumors. They analyzed 203 unilateral multifocal WTs and compared 96 patients managed by multifocal extracapsular dissection to 107 patients managed by total parotidectomy. The authors found metachronous tumors in 7 patients (3.4%) out of the 96 patients reviewed, with a significant difference in the occurrence rate between multifocal ECD (7/96, 7.3%) compared to total parotidectomy (0/107, 0%) (*p* = 0.004). Permanent facial palsy was significantly more common after total parotidectomy (21/107, 19.6%) than in multifocal extracapsular dissection (2/96, 2.1%) (*p* = 0.000). The authors concluded that the functional benefit of multifocal ECD was counterbalanced by the advantage of total parotidectomy in terms of second WTs.

Even when multifocal ECD is possible, we believe this option must be employed only for selected patients and by dedicated surgeons with experience in multifocal ECD. The consensus (Table 2) for multiple tumors involving superficial and deep lobe lesions should be a complete parotidectomy.

### 7.2. Parotidectomy I–II (Lateral Parotidectomy)

As most WTs are lateral to the facial nerve, the most common strategy has been lateral (or superficial) parotidectomy. Therefore, different authors presented it as the most common surgical approach performed in WTs [9,36]. Additionally, in case of multifocal lesions, parotidectomy I–II (with a careful intraoperative exploration of the entire parenchyma) should be an appropriate surgical option [34]. This is also the preferred option (Table 1) for multiple WTs that involve levels I and II, or in case of a larger WT (category III), that involves superficial levels.

### 7.3. Parotidectomy II (Partial Lateral Parotidectomy)

As the majority of WTs are situated in the level II (parotid tail), parotidectomy II, or partial lateral (or superficial) parotidectomy, is the most commonly performed less extensive surgical procedure instead of lateral parotidectomy. Leverstein et al. [37] stressed the need to reduce surgical morbidity with less invasive surgical modalities for WTs, as the risk of a second microscopic lesion in the remaining gland becoming clinically manifested is considered extremely low. Many authors follow this strategy but combine different extensions depending on the extent of the tumor. For example, Lee et al. [33] in a recent series of 118 WTs treated with surgery, performed a partial parotidectomy in 43 cases. The rest were treated with a superficial parotidectomy (68 cases) or a total parotidectomy (7 cases). This partial parotidectomy II (Table 2) is indicated for small WTs (categories I or II) or for larger WTs originating in level II but extending to the upper neck (Table 2).

### 7.4. Extracapsular Dissection (ECD)

In recent years, many authors have favored ECD in WTs. Witt el al. [31] reviewed 50 consecutive WTs, half of them treated with partial superficial parotidectomy and the other half with ECD with satisfactory results. They concluded that minimally invasive approaches can be performed with confidence with ECD, resulting in potentially reduced morbidity. Different authors [38,39] have also reported satisfactory results with ECD. Mantsopoulos et al. [40] analyzed 327 WTs treated with ECD, observing an overall occurrence rate of 3.1% for ipsilateral second metachronous tumors. They concluded that ECD is extremely useful in the management of solitary as well as carefully selected multicentric WTs with acceptable surgical outcomes.

However, ECD has generated some controversies, as it is a less standardized technique and long-term data are less extensive than with standard techniques. As mentioned, one must keep in mind that [6] the incomplete excision of WTs is clearly related to a high risk of recurrence. In contrast, the intraoperative rupture of a WT has not been associated with tumor recurrence [6]. ECD is regarded as a good option mainly for solitary WTs of category I and II (Table 2).

### 7.5. Capsular Dissection (CD)

An even more conservative surgical procedure than ECD for WTs has been proposed by Mantopoulos et al. [41] These authors proposed the term capsular dissection (CD) and defined it as the removal of a WT around its capsule without any margin of healthy tissue around it. They compared 34 CDs with 595 ECDs, detecting a 2.06% (13/629) overall occurrence rate of ipsilateral metachronous tumors all after ECD, with a mean follow-up of 104.1 months. No statistical difference could be found between ECD and CD in the overall occurrence of ipsilateral metachronous tumors or in the complication rate. In fact, McMullen et al. [42] in 2016 proposed the term minimal margin extracapsular dissection (MECD) with dissection on or closer to the tumor capsule as an alternative to ECD. CD and MECD can have theoretically similar indications as ECD in WTs but warranting a complete resection of WT. However, since the series are still limited in size, we believe that those techniques should be applied in centers with experienced surgical teams and with strict follow-up protocols for these patients.

## 8. Active Surveillance (AS)

WTs usually occur in patients over 50 years of age, many WTs have been found incidentally, and they are often asymptomatic. The risk of malignant transformation is extremely low, and the growth rate can be slow. Thus, some authors propose AS as the first step for those tumors [7,43,44]. Additionally, some patients may have comorbidities that make this option more appropriate. AS is particularly a good option for “incidentalomas”, in elderly patients, or patients with significant comorbidities, and without clinical complains. Since smoking is clearly associated with WTs, many WT patients will have comorbidities related to smoking, i.e., cardiovascular and respiratory diseases.

There is a trend to recommend 6 months of clinical follow-up with ultrasonographic evaluation to assess interval growth. The advantage of AS is to avoid surgery considering that some tumors remain asymptomatic over a long period of time, and some may even shrink. There have been reports of partial or complete resolution of a WT following fine needle aspiration cytology [22,43,45].

The standard advice for smokers should be smoking cessation. This will benefit this population because continued smoking increases the risk of metachronous WTs [46].

The main drawback of AS is the potential variability in the behavior of a WT: some remain asymptomatic, whereas others may grow in a short period of time. Patients with tumors that grow progressively will eventually have to undergo more extensive surgeries. Schwalje et al. studied the growth rate of WTs [47] and found a high variability, with an average volume doubling in approximately 9 years, while some lesions doubled within a year. Age over 75 years was associated with slower growth, but gender, smoking status, multifocality, bilaterality versus unilaterality, and initial size did not correlate with the growth rate. In addition, the authors found many cases showing a size reduction. Another study [44] on metachronous WTs confirmed the growth rate variability with a mean rate of growth per lesion of 15%/year.

Currently, the AS approach should be considered in all cases for which immediate surgery is not indicated (Figure 3).

## 9. Alternatives to Surgery and AS

Experience with non-surgical options in the management of WTs is limited. Three recent studies described three potential new approaches: microwave ablation (MWA) [48], radiofrequency (RF) [49], and ultrasound-guided ethanol sclerotherapy (UGES) [50]. The first two options have a long history in the treatment of other neoplasms, especially liver tumors, with a therapeutic effect based on thermal injury. MWA uses an electromagnetic field as opposed to the electric current used in RF. UGES is commonly used in vascular malformations and has been effective in reducing the size of benign thyroid nodules, branchial cleft cysts, and parotid cysts. Ethanol induces coagulation necrosis through cellular dehydration and protein denaturation, ultimately resulting in shrinkage of the lesion.

### 9.1. Microwave Ablation (MWA)

One recent publication [48] presented a case of ultrasound guided percutaneous MWA of a WT. The ablation was a short and easy procedure without any adverse effects. The reduction ratios of the tumor volume at the 1- and 3-month follow-up were 53% and 82%, respectively, and the remnant finally disappeared at 8 months with no recurrence at 2 years. Thus, the authors pointed out that ultrasound guided percutaneous MWA seems to be a safe and effective treatment for parotid gland WTs, and it may serve as a novel alternative approach for patients unsuitable for open surgery.

### 9.2. Radiofrequency Ablation (RFA)

Another recent publication [49] analyzed seven consecutive patients with biopsy-proven parotid WTs who were treated with US-guided percutaneous RFA ablation under local anesthesia. All procedures were uneventful with no complications. One patient had parotitis with hematoma 12 days after parotid RFA. There was no facial nerve injury. All the tumors were significantly reduced, achieving a 94.3% ± 2.0% of volume reduction at 10–12 months after RFA.

### 9.3. Ultrasound-Guided Ethanol Sclerotherapy (UGES)

Recently, Mamidi et al. [50] published the first study of UGES for management of WTs. They treated two patients with a total of three WTs. Both patients experienced a significant reduction in tumor size during the follow-up (volume of tumor reduction ranging from 55.73% to 98.32% upon post-procedural evaluation). Patients were satisfied with the results and noted significant cosmetic improvement. No complications were observed at follow-up. UGES may be a viable option for conservative treatment of WT in patients unsuitable or unwilling to undergo surgical resection. In comparison to MVA and RFA therapy, UGES is considered less painful. UGES has been widely used in otolaryngology as a nonsurgical treatment method for cystic benign thyroid nodules, branchial cleft cysts, and thyroglossal duct cysts. The sclerosing effect of ethanol has been particularly efficacious in a partial and total reduction of lymphatic and venous malformations of the head and neck. Ethanol ablation treatment does not eradicate the WT completely; instead, it induces the regression of it.

These three non-surgical options may have good potential, as demonstrated in the management of other neoplasms, and may change the management of many of these tumors in the future. However, more evidence with long-term prospective outcome studies is needed to establish their proper role before widespread use.

## 10. Conclusions

From this review, it appears that the incidence of WT diagnosis has progressively increased over the last decades in Europe, without a clear cause. In some countries, it has become the most frequent benign tumor of the parotid gland. This high incidence stresses the importance of discussing the controversies in its management. A reliable diagnosis is based on concordant clinical, imaging, and cytology data. In doubtful cases, surgical resection is advised to obtain a definitive histopathological diagnosis. If the diagnosis of a WT is proven, upfront surgery is indicated in cases of cosmetic or clinical complaints, or the patient′s wish, if medically reasonable. In the remaining cases, active observation is an option and should be offered to the patient.

When the surgical option is chosen, the extent of surgery must be personalized. Category I or II (less than 3 cm) tumors probably are best treated with conservative resections. In larger tumors, most authors favor more extensive resections, including superficial parotidectomies or even total parotidectomies when the deep lobe is involved. In selected cases, alternative non-surgical treatments are emerging, i.e., microwave ablation or radiofrequency ablation and ultrasound-guided ethanol sclerotherapy, which may play an important role in the treatment of WTs in the future. More studies are needed to establish the role of these treatment modalities.

## Figures and Tables

**Figure 1 diagnostics-11-01467-f001:**
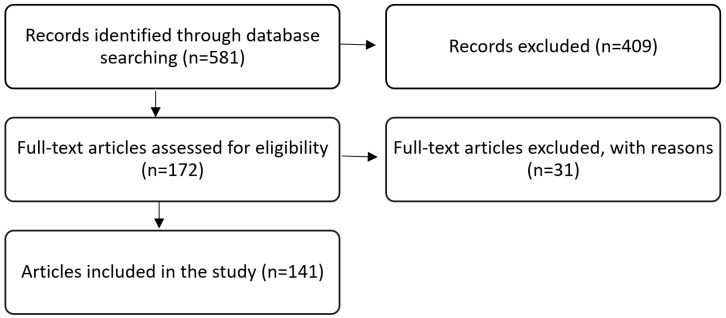
Flow chart showing the process of the study selection for the systematic review.

**Figure 2 diagnostics-11-01467-f002:**
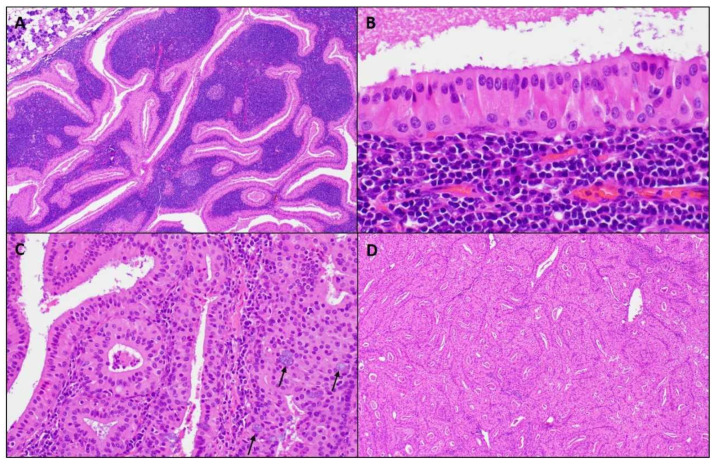
Microscopic images of a WT. (**A**) Well-circumscribed parotid tumor with variable papillary and cystic architecture composed of oncocytic epithelial elements and a prominent lymphoid stroma, (magnifications, 4 × 10); (**B**) The oncocytic epithelium shows a bilayer of inner luminal tall cells with nuclei polarized towards the lumen and outer cuboidal basal cells with basally located nuclei, (magnifications, 40 × 10); (**C**) Scattered mucocytes (arrows) can be appreciated in the oncocytic epithelium, (magnifications, 20 × 10); (**D**) Lymphoid stroma-poor WT, (magnifications, 4 × 10).

**Figure 3 diagnostics-11-01467-f003:**
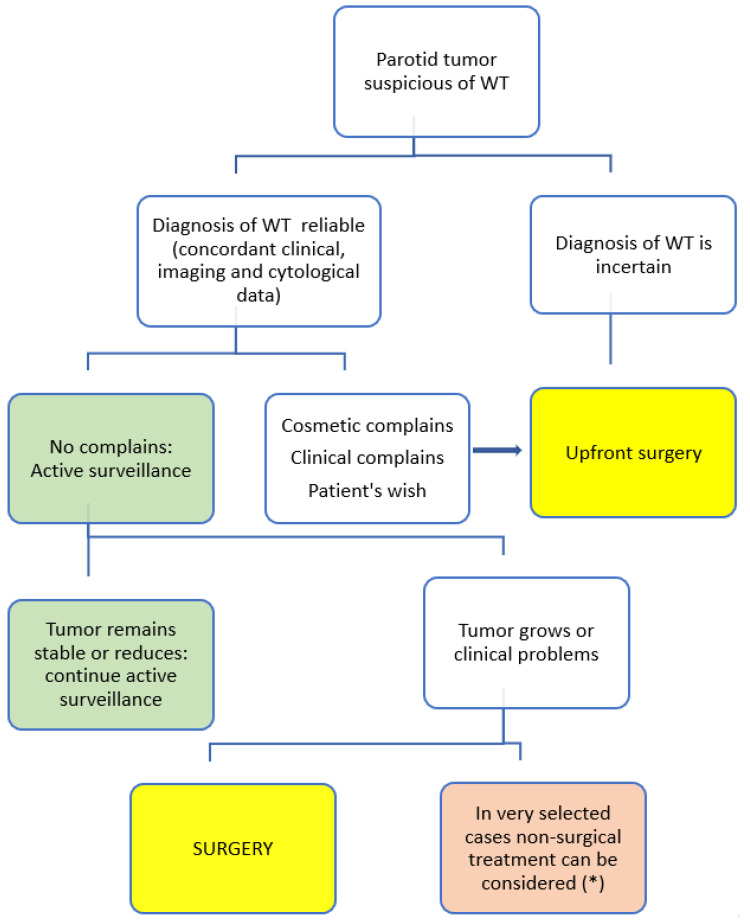
Proposed treatment guideline. (*) Non-surgical treatments used until now in WT are microwave ablation, radiofrequency ablation, and ethanol sclerotherapy.

**Figure 4 diagnostics-11-01467-f004:**
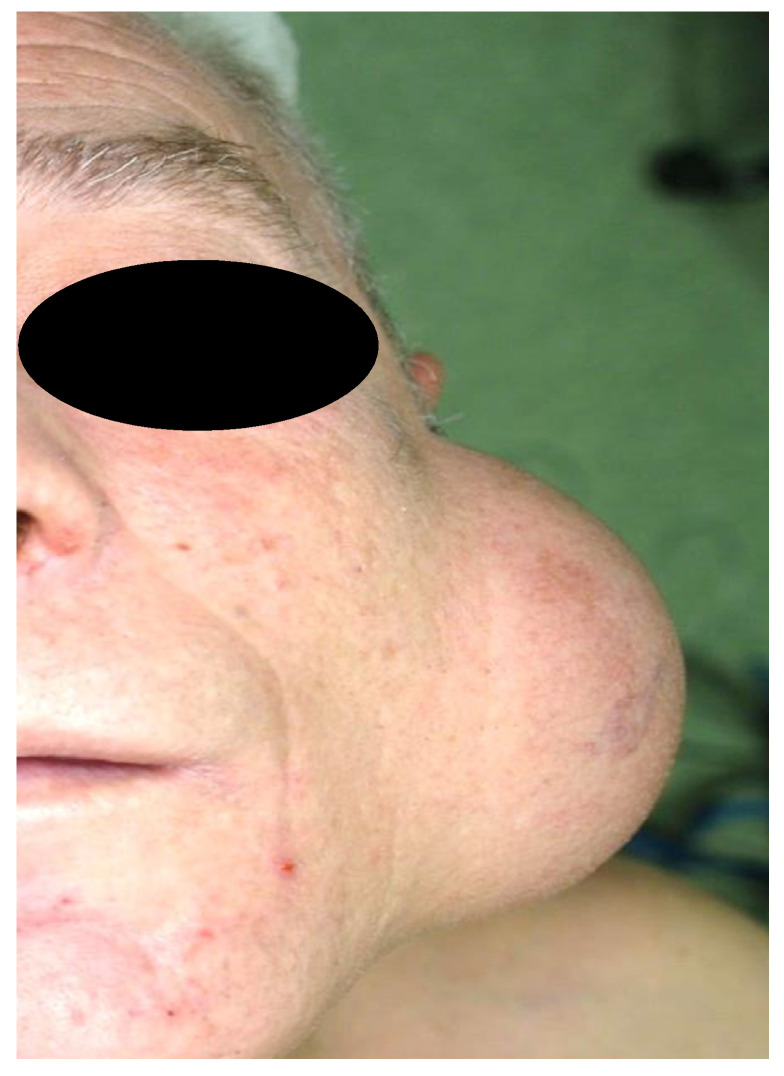
A typical large WT causing cosmetic problems.

**Figure 5 diagnostics-11-01467-f005:**
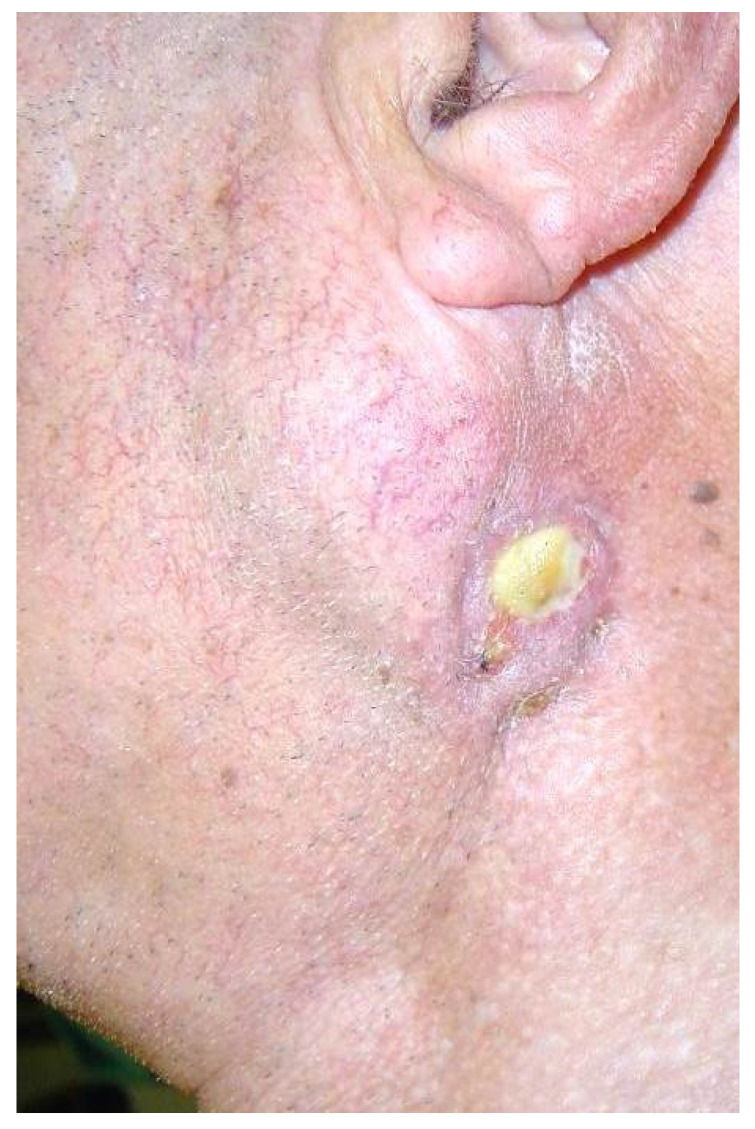
WT presented with ulceration is also an indication for upfront surgery.

**Table 1 diagnostics-11-01467-t001:** Indications for upfront surgical approach.

Clinical Scenarios	Comments
1. Diagnosis of WT is not reliable	A reliable diagnosis must be based on concordant clinical, image, and cytological data
2. Cosmetic concerns	When the WT grows considerably causing cosmetic complaints, surgery is the best option
3. Clinical complaints: pain, ulceration, or recurrent infection	When the WT is painful or ulcerates, surgery is advised.
4. Patient’s wishes	

**Table 2 diagnostics-11-01467-t002:** Proposed surgical approach in various clinical settings.

WT	Resections Proposed
Only one lesion category I or II (*)	Parotidectomy II (partial parotidectomy) or ECD
Only one lesion category III or IV (*) intraparotid	Parotidectomy adapted to the extension of the lesions
Only one lesion of more than 3 cm in parotid tail, but extending mainly to the extraparotid	Parotidectomy II (partial lateral parotidectomy) or ECD
Multiple lesions affecting superficial parotid	Parotidectomy I and II (lateral or superficial parotidectomy)
Multiple lesions affecting superficial and deep lobe	Parotidectomy I–II–III–IV (total parotidectomy)

(*) Categories defined by Quer et al. [24].

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
