# Peer review of "Current Trends and Controversies in the Management of Warthin Tumor of the Parotid Glandâ€"

_diagnostics, 2021, doi:10.3390/diagnostics11081467_

Round 1
Reviewer 1 Report
This manuscript is a comprehensive international review on Current Trends and Controversies in the Management of Warthin Tumor of the Parotid Gland.
It is a very interesting work, thus I strongly recommend its publication.
However, I suggest some minor changes:
Line 99: The author state “A scoping review of the current literature was conducted using PubMed and ScienceDirect databases from 1980 to 2021, last update on May 18, 2021, for English language”.
I have received this manuscript in July 24th, two months later.
I suggest them to include also the following very recent paper:
Mashrah MA, Al-Sharani HM, Al-Aroomi MA, Abdelrehem A, Aldhohrah T, Wang L. Surgical interventions for management of benign parotid tumors: A systematic review and network meta-analysis. Head Neck. 2021 Jul 20. doi: 10.1002/hed.26813.
Line 411: It should be corrected. The authors refer to Figure 3 (algorithm).
Line 388: The authors state: “Thus, many authors propose active surveillance as the first step for those tumors (7,43,44)”.
I think this sentence should be changed. Many authors? Some authors?
Only three references state that active surveillance is proposed as the first step. Reference 44 states “The extreme variability in the behaviour of metachronous cystadenolymphomas points to the need for an individualised approach, accurate investigation of the sonographic characteristics, and continuous watchfulwaiting in affected patients.”
This would be more consistent to line 274 of this paper: “Surgery currently remains the primary recommended treatment of WT. The extent of the surgery depends on the size and location of the tumor (28,29), and line 462: “In the remaining cases, active observation is an option and should be offered to the patient.”
Author Response
RESPONSE TO REVIEWER 1 COMMENTS
Point 1: It is a very interesting work, thus I strongly recommend its publication.
Response 1: Thank you very much for your work reviewing the manuscript and your positive comment.
Point 2: However, I suggest some minor changes: Line 99: The author state “A scoping review of the current literature was conducted using PubMed and ScienceDirect databases from 1980 to 2021, last update on May 18, 2021, for English language”. I have received this manuscript in July 24th, two months later. I suggest them to include also the following very recent paper: Mashrah MA, Al-Sharani HM, Al-Aroomi MA, Abdelrehem A, Aldhohrah T, Wang L. Surgical interventions for management of benign parotid tumors: A systematic review and network meta-analysis. Head Neck. 2021 Jul 20. doi: 10.1002/hed.26813.
Response 2: Thank you very much for your very pertinent comment. However, we set a final review date for several reasons: it takes time to complete the review of the articles, plus we have to establish consensus among the authors and finally reach agreement on the final manuscript. For this reason, there is always a few months between the end point of the review and the submission of the manuscript.
Although the publication suggested by you is very interesting, we believe that adding only one more recent citation is unjustified. There are other interesting publications that appeared after May 18, 2021, and we believe that the most correct thing to do is precisely to keep the final review date so that readers are clear that there will be new publications after this date that have not been included.
Point 3. Line 411: It should be corrected. The authors refer to Figure 3 (algorithm).
Response 3: Thank you very much, you are right. It has been corrected in the new text.
Point 4. Line 388: The authors state: “Thus, many authors propose active surveillance as the first step for those tumors (7,43,44)”. I think this sentence should be changed. Many authors? Some authors?
Response 4: Yes, probably "some authors" is more appropriate. We have corrected it in the new text.
Point 5. Only three references state that active surveillance is proposed as the first step. Reference 44 states “The extreme variability in the behaviour of metachronous cystadenolymphomas points to the need for an individualised approach, accurate investigation of the sonographic characteristics, and continuous watchfulwaiting in affected patients.” This would be more consistent to line 274 of this paper: “Surgery currently remains the primary recommended treatment of WT. The extent of the surgery depends on the size and location of the tumor (28,29), and line 462: “In the remaining cases, active observation is an option and should be offered to the patient.”
Response 5. Yes, thank you very much again for your comment. It is true that our text reflects in some way the contradictions that exist at the present time regarding the treatment of these tumors. We believe that individual assessment is fundamental to decide in each case what best suits the patient's situation, and we hope that this is reflected in Figure 3, where we try to give a role to each option.
Reviewer 2 Report
The manuscript very clearly summarizes the possibilities of diagnosis and subsequent treatment of Warthin's tumor of the parotid gland. It shows how the topic is conceived differently and controversially across publications. But in my opinion, it is necessary to mention (discuss) that in Whartin's tumor, it is possible to choose only monitoring without the need for treatment.
Author Response
RESPONSE TO REVIEWER 2 COMMENTS
Point 1: The manuscript very clearly summarizes the possibilities of diagnosis and subsequent treatment of Warthin's tumor of the parotid gland. It shows how the topic is conceived differently and controversially across publications.
Response 1: Thank you very much for your work reviewing the manuscript and your positive comments.
Point 2: But in my opinion, it is necessary to mention (discuss) that in Whartin's tumor, it is possible to choose only monitoring without the need for treatment.
Response 2: We fully agree with you about not treating these tumors in a lot of cases. In fact, the entire section on "active surveillance" (between lines 385 to 412) discusses in deep this option.
Also, in the conclusions (lines 460 to 463) we are very insistent on this point "In the remaining cases, active observation is an option and should be offered to the patient". We believe that such a conclusion is in full agreement with your comment.